# Protein Binder (ProBi) as a New Class of Structurally Robust Non-Antibody Protein Scaffold for Directed Evolution

**DOI:** 10.3390/v13020190

**Published:** 2021-01-27

**Authors:** Phuong Ngoc Pham, Maroš Huličiak, Lada Biedermannová, Jiří Černý, Tatsiana Charnavets, Gustavo Fuertes, Štěpán Herynek, Lucie Kolářová, Petr Kolenko, Jiří Pavlíček, Jiří Zahradník, Pavel Mikulecky, Bohdan Schneider

**Affiliations:** Institute of Biotechnology of the Czech Academy of Sciences, BIOCEV, CZ-25250 Vestec, Czech Republic; phuong@ibt.cas.cz (P.N.P.); maros.huliciak@ibt.cas.cz (M.H.); Lada.Biedermannova@ibt.cas.cz (L.B.); jiri.cerny@ibt.cas.cz (J.Č.); tatsiana.charnavets@ibt.cas.cz (T.C.); gustavo.fuertes@ibt.cas.cz (G.F.); Stepan.Herynek@ibt.cas.cz (Š.H.); kolarovalucie007@gmail.com (L.K.); kolenko@ibt.cas.cz (P.K.); jiri.pavlicek@ibt.cas.cz (J.P.); jiri.zahradnik@ibt.cas.cz (J.Z.)

**Keywords:** directed evolution, protein scaffold, protein engineering, computational saturation, ribosome display, interleukin-10

## Abstract

Engineered small non-antibody protein scaffolds are a promising alternative to antibodies and are especially attractive for use in protein therapeutics and diagnostics. The advantages include smaller size and a more robust, single-domain structural framework with a defined binding surface amenable to mutation. This calls for a more systematic approach in designing new scaffolds suitable for use in one or more methods of directed evolution. We hereby describe a process based on an analysis of protein structures from the Protein Data Bank and their experimental examination. The candidate protein scaffolds were subjected to a thorough screening including computational evaluation of the mutability, and experimental determination of their expression yield in *E. coli*, solubility, and thermostability. In the next step, we examined several variants of the candidate scaffolds including their wild types and alanine mutants. We proved the applicability of this systematic procedure by selecting a monomeric single-domain human protein with a fold different from previously known scaffolds. The newly developed scaffold, called ProBi (Protein Binder), contains two independently mutable surface patches. We demonstrated its functionality by training it as a binder against human interleukin-10, a medically important cytokine. The procedure yielded scaffold-related variants with nanomolar affinity.

## 1. Introduction

Many biological functions depend on specific protein-protein interactions. Protein engineering offers the possibility to tune these interactions by developing de novo binding partners [1,2,3] or by mutating the interaction partners using computational design [4,5]. A powerful tool of protein engineering is to generate protein binders by the in vitro directed selection techniques [6,7,8] or to use evolution-based approaches to increase the stability of recombinant proteins [9,10,11].

A prototype of a binding protein is an antibody, a highly selective and adaptive molecule capable of binding to a huge spectrum of partners. The antibody-based binders have been indispensable not only in research experiments but also in clinical trials and tens of them are biologicals approved by Federal Drug Administration. However, several suboptimal properties of these molecules such as their large size (~150 kDa), cross-reactivity, and necessity of animal immunization during the preparation, motivated the development of binders with alternative structures [12,13]. These novel artificial high-affinity binders are called “small non-antibody protein scaffolds” [14,15,16]. The advantage of scaffolds over antibodies is their typically higher stability and lower cost of production. The most widely used scaffolds are designed Ankyrin repeats (dARPins), lipocalin domain (anticalins), and Ig-binding domain of Staphylococcal Protein A (affibodies) [17,18,19]. Some of the small protein scaffolds were already proven to be useful in a wide range of applications, from academic research to clinical imaging, diagnostics, or therapeutics [20,21,22,23], and others are even in clinical trials [24,25]. In our previous work, we used the albumin-binding domain (ABD) of Streptococcal Protein G as a scaffold protein. We engineered its binding against human IFNγ [26] and interleukin-23 receptor [27] using computational design and ribosome display.

The exploration of non-antibody protein scaffolds is still a young field compared to the research of antibodies and the full potential of scaffolds has not yet been uncovered. Therefore, we are convinced that there is ample space to expand the current portfolio of protein scaffolds so that they can complement antibodies and interact with new target molecules. Therefore, a systematic method to streamline the development of new scaffolds with desired properties would be beneficial but relatively few attempts can be found in the literature [18,28,29]. In general, a good small protein scaffold should display a high stability, together with flexibility of binding. The desired properties thus include tolerance to mutations during diversification, thermostability, in vivo integrity, and non-immunogenicity. Moreover, the variants of the protein scaffold must be efficiently evolvable to achieve high affinity and specificity towards target proteins. Therefore, the properties of the potential small protein scaffold must be thoroughly evaluated.

In this work, we present a systematic stepwise procedure for the selection of new and structurally robust protein scaffolds. The procedure starts with a wide search of suitable structures in the Protein Data Bank (PDB) [30], followed by a series of computational and experimental screens to select the most robust candidates as novel, small protein scaffolds. We demonstrate the applicability of our method by selecting a protein scaffold called ProBi (Protein Binder) that fulfills all the criteria of stability and mutability, and comprises not one, but two independently mutable surface patches.

We tested the applicability of the ProBi scaffold by generating its variants with high affinity to a target protein using in vitro directed evolution techniques. As the target, we selected a medically important protein, interleukin-10 (IL-10), a member of the cytokine superfamily. IL-10 is an immune repressor [31], with numerous investigations suggesting its major impact in inflammatory, malignant, and autoimmune diseases. IL-10 overexpression was found in certain tumors and is considered to promote tumor development [32]. Also, some viral homologs can bind to the IL-10 receptor 1 [33] and mimic the immunosuppressive effects of IL-10. Thus, proteins binding IL-10 and interfering with its signaling have a great potential for diagnostic and therapeutic purposes.

To generate ProBi-based binders to IL-10, we created a DNA library by randomization of 10 amino acids on one of the ProBi surface patches and used the ribosome display technique to screen for ProBi variants with the highest affinity. We successfully generated binders of the newly developed ProBi protein scaffold with affinities of ~10 and ~200 nM against IL-10.

## 2. Materials and Methods

### 2.1. Selection of Suitable Protein Structures from the Protein Data Bank

The Protein Data Bank (PDB) [30] was searched for potential protein scaffolds using the following criteria: (1) small size (molecular weight in range of 10–25 kDa); (2) monomeric protein with a reasonably high-resolution X-ray structure (<3.0 Å); and (3) produced in Expression Organism of *E. coli* for ease of expression. Proteins matching these initial search criteria were then manually curated to meet additional rules: structure (protein fold class) (1) should differ from the previously published protein scaffolds; (2) should differ from the other selected structures in the set; (3) should contain a low number of cysteine residues and disulfide bonds, (4) should not be toxic or immunogenic, (5) should not contain any ligand or cofactor, and (6) should be soluble and easy to purify (according to the source literature).

### 2.2. In Silico Identification of Mutable Surface Patches on Protein Scaffolds Candidates

(a) Multiple sequence alignment. For each of the scaffold candidate proteins, a similarity search was performed for the related amino acid sequences from different species using the UniProt BLAST service (www.uniprot.org/blast) with default parameters (Target database = UniProtKB, E-Threshold = 10, Matrix = blosum62, Filtering = None, Gapped = yes, Hits = 250). The identical sequences were subsequently removed using a script and the remaining ones were aligned using Clustal W [34] in the UGENE program [35]. The conservation of each sequence position was calculated.

(b) In silico saturation mutagenesis. The mutability of the surface amino acid residues was evaluated by in silico mutation scanning using the FoldX program [36] by applying the “positionscan” FoldX Keywords: In all proteins, each position was substituted by all of the 20 standard amino acid residues, and the corresponding free energy differences (ΔΔG) between the mutant and the wild-type (WT) structure were calculated; the calculations included self-mutations leading to ΔΔG = 0. Positions in which most of the mutations were stabilizing (ΔΔG < 0), or only slightly destabilizing (ΔΔG < 0.5 kcal/mol), were identified and the mutability score for each position was calculated as a percentage of mutations fulfilling these criteria.

(c) Sequence and structural evaluation. The mutable residues, identified by multiple sequence alignment and in silico saturation mutagenesis, were located on the 3D structure of each protein scaffold candidate. We then visually searched for continuous surface regions consisting of mutable residues. The regions where the mutable residues were close together in both sequential and structural space we termed the mutable surface patch. Ideally, the patch should contain 10–12 residues to fulfill the theoretical complexity suitable for ribosome display. If the identified patch contained less than 10–12 residues with the highest mutability score, we completed the patch by a small number of neighboring surface residues with slightly lower mutability scores.

### 2.3. Production of Proteins

(a) Cloning of recombinant proteins. The genes coding the selected five protein scaffold candidates–4PSF, 1N3Y, 4I3B, 1W2I, and 2F3L, were ordered in the form of DNA strings (ThermoFisher Scientific, Waltham, MA, USA) with codons optimized for expression in *Escherichia coli* but without a stop codon. The DNA strings were cloned into the pET-26b(+) vector using NdeI and XhoI restriction enzymes in frame with C-terminal His-tag. Also, the DNA strings were used as templates for polymerase chain reaction (PCR) to add the N-terminal His-tag and stop codon. These N-terminally tagged proteins were cloned into the pET-26b(+) vector using NdeI and XhoI restriction enzymes. Final constructs were labeled as pET26b-4PSF-WT, etc.

The genes coding the variants of three protein scaffold candidates–4PSF, 1N3Y, and 4I3B, with all mutable residues changed to alanine amino acid, referred to as allAla mutants, were ordered in the form of DNA strings with N-terminal His-tag and C-terminal stop codon. The DNA strings were digested by NdeI and XhoI restriction enzymes and cloned into the pET-26b(+) vector. Final constructs were labeled as pET26b-4PSF-allAla, etc.

The ProBi-WT protein and its variants were amplified by the PCR method using the ProBi-cloning-for and ProBi-cloning-rev primers (Appendix A). The created fragments were cleaved by NcoI and BamHI restriction enzymes and cloned into the pRDVsm or pETsm vectors (Appendix A).

The genes encoding the interleukin-10 (residues 19–179 of UniProt entry P22301) and interleukin-29 (residues 20–200 of UniProt entry Q8IU54) were ordered in the form of a DNA string with codons optimized for expression in *Drosophila melanogaster*. The DNA was cleaved by BglII and XhoI restriction enzymes and cloned into a pMTH vector (a modified version of pMT-BiP-V5-His_A vector where recognition site for the AgeI restriction enzyme is exchanged for XhoI enzyme). The final construct contained IL-10 or IL-29 protein with N-terminal BiP signal peptide and C-terminal His-tag. Final constructs were labeled as pMTH-IL10cd and pMTH-IL29cd.

(b) Expression and purification of recombinant proteins. The recombinant protein scaffold candidates were produced in *E. coli* BL21(DE3) strain. In addition, the expression of 1W3I scaffold was tested in *E. coli* C41(DE3) and C43(DE3) strains. The bacteria were cultivated in LB medium (Sigma-Aldrich, St. Louis, MO, USA) at 37 °C until OD600 = 0.6, followed by the addition of 1 mM isopropyl-beta-D-thiogalactopyranoside (IPTG) for induction of expression. Then the cultivation continued for 4 h at 37 and 30 °C or overnight at 25 and 16 °C. The cells were harvested by centrifugation (5000× *g*, 15 min, 4 °C) and stored at −20 °C. The cells were resuspended in LH buffer (20 mM Na-Phosphate buffer, pH 7.3, 100 mM NaCl), and disrupted by ultrasound. The soluble fraction was separated by centrifugation (40,000× *g*, 20 min, 4 °C) and passed over the Ni-NTA agarose beads equilibrated in LH buffer. The beads were washed by LH buffer supplemented by 10 mM imidazole and proteins were eluted by LH buffer containing 200 mM imidazole.

The recombinant ProBi variants were produced in *E. coli* BL21-Gold(DE3) in LB medium (Sigma-Aldrich) at 37 °C until OD600 = 0.6, followed by the addition of 1 mM IPTG for induction of expression. Then cultivation continued overnight at 18 °C. The cells were harvested by centrifugation (5000× *g*, 15 min, 4 °C) and stored at −20 °C. The cells were resuspended in LS buffer (50 mM Tris, pH 8.0, 150 mM NaCl, 1 mM EDTA), and disrupted by ultrasound. The soluble fraction was separated by centrifugation (40,000× *g*, 20 min, 4 °C) and passed over the StrepTactin XT resin equilibrated in LS buffer. The resin was washed by LS buffer and proteins were eluted by BXT buffer (100 mM Tris, pH 8.0, 150 mM NaCl, 1 mM EDTA, 50 mM biotin).

The constructs pMTH-IL10cd and pMTH-IL29cd for expression of recombinant IL-10 and IL-29, respectively, in insect S2 cells were produced and purified similarly as IFNgR2 in the previous report [37]. Briefly, S2 cells were cultivated in Insect-XPRESSTM Protein-free Insect Cell Medium (Lonza) and protein expression was induced by the addition of CuSO4. IL-10 and IL-29 proteins were secreted into the medium and stored at −80 °C.

All proteins were further purified to homogeneity by size exclusion chromatography at 16 °C using Superdex 75 10/300 Increase, Superdex 200 10/300 Increase, or Superdex 75 16/600 column (GE Healthcare, Chicago, IL, USA). The column was equilibrated in phosphate-buffered saline (PBS) pH 7.4 (for protein scaffold proteins and all Ala mutants), 20 mM Tris, pH 7.5, 100 NaCl (for preliminary crystallization trials), or in 50 mM Tris, pH 8.0, 300 mM NaCl (for ProBi variants and IL-10 protein). Samples were analyzed by 15% sodium dodecyl sulphate–polyacrylamide gel electrophoresis (SDS-PAGE).

### 2.4. Biophysical Characterization of Recombinant Proteins

(a) Oligomerization and melting temperature of protein scaffold candidates and allAla mutants. The oligomeric state of proteins and their monodispersity was measured by dynamic light scattering (DLS) using a ZetaSizer Nano ZS90 (Malvern Panalytical, Malvern, UK) instrument and quartz cuvette ZEN2112. The secondary structure and folding of proteins were analyzed by circular dichroism (CD) spectra using a Chirascan Plus spectrometer (Applied Photophysics, Surrey, UK) in steps of 1 mm over the wavelength range 200–260 nm. Samples were diluted with PBS buffer, pH 7.4, to a concentration of 0.2 mg/mL. The individual spectra were measured in a 0.01 cm path-length quartz cell at a temperature of 25 °C. The CD signal was recorded as the ellipticity, and the resulting spectra were buffer-subtracted. The melting temperatures of proteins were evaluated using either the CD spectrometer or nanoDSF technology. The CD melting measurements were performed by using the 0.1 cm path-length quartz cell. The samples were heated from 20 to 85 °C, and the sample absorption was recorded at 280 nm in 1 °C increments at a rate of 0.5 °C/min. The nanoDSF technology was implemented in the Prometheus NT.48 instrument (NanoTemper, München, Germany). The samples were loaded into standard capillaries and heated from 20 to 95 °C at a rate of 1 °C/min. The melting temperatures (Tm) were estimated from the first derivative of the melting curves.

(b) Testing protein suitability for affinity measurements. The applicability of protein scaffold candidates and allAla mutants for affinity measurements using the microscale thermophoresis (MST) and surface plasmon resonance (SPR) was tested. For MST, proteins were labeled using the Monolith Protein Labeling Kit RED-NHS (Amine Reactive) according to the manufacturer’s protocol. The samples were loaded into three types of capillary (standard, hydrophilic, and hydrophobic) and inserted into the Monolith NT.115 instrument (NanoTemper). The capillary scan was performed immediately, and then samples were incubated in the instrument for 1 h and re-measured. An alternative approach was implemented for SPR. The GLC sensor chip was inserted into the ProteOn XPR36 instrument, washed but not activated. The samples in different channels were passed over the un-activated chip to estimate the non-specific binding of proteins to the surface of the chip itself. The running buffer (PBS, pH 7.4, 0.005% Tween-20) was used in the control channel.

### 2.5. Crystallization and Structure Determination

The crystallization solution included J61 at a concentration of 10 mg/mL in a buffer containing 100 mM NaCl, 20 mM Tris, pH 8.0. Crystals were prepared using the hanging-drop vapor-diffusion method with the reservoir solution containing 5% (*v*/*v*) glycerol and 4 M sodium formate. Crystals required cryo-protection with 20% glycerol before flash freezing in liquid nitrogen and the diffraction experiment.

The final diffraction data were collected at the Helmholz–Zentrum Berlin (Bessy II) on a beamline 14.1 at 100 K. The diffraction images were processed using the XDS program [38] and scaled using the AIMLESS program [39] from the CCP4 program package [40]. The phase problem was solved by molecular replacement using PHASER [41] using monomeric 4PSF structure [42] as a search model. The structure was refined using the REFMAC5 program [43] using 95% of reflections as a working set and 5% or reflections as a test set. Manual corrections to the model were done using the graphic COOT program [44]. The last run of several cycles of refinement was performed using all reflections and anisotropic ADPs. The structure quality was checked using the validation tools implemented in CCP4, MOLPROBITY [45], and the PDB. Final data processing statistics and structure refinement statistics are shown in Appendix A.

### 2.6. Construction of DNA Library of ProBi Scaffold (PatchC)

The earlier design of the DNA library [46] was adopted with small changes regarding restriction enzyme recognition sites and N- and C-terminal tags. Our ribosome display DNA construct contained a T7 promoter, 5′ stem-loop, ribosome binding site, N-terminal Strep-tag, open reading frame encoding the ProBi scaffold library, and C-terminal c-Myc-tag fused to TolA-spacer without stop codon. The library of ProBi scaffold (PatchC) was prepared by the GENEWIZ company using NNK codons technology. We selected ten residues to be randomized. The initial synthetic DNA library was amplified by PCR and then used in the first selection round of ribosome display.

### 2.7. Selection of Novel Binders by In Vitro Ribosome Display

The previous protocols from Pluckthun’s laboratory regarding the in vitro selection by ribosome display [46,47] were adopted and adjusted. All the following steps were first performed with the ProBi-WT scaffold to establish and verify the protocol.

(a) Preparation of the mRNA-ribosome-protein (MRP) complex. The DNA library was amplified by PCR (supplemented by 6% DMSO) with primers T7b and TolAk (Appendix A) using Q5 polymerase (New England Biolabs, Ipswich, MA, USA) with an annealing temperature of 66 °C. As a template, either the starting DNA library in the first selection round was used or the sorted DNA library cloned into the pRDVsm vector in the next rounds. The resulted amplicon contained a T7 promoter, 5′ stem-loop, ribosome binding site (RBS), start codon, N-terminal Strep-tag II, scaffold library, C-terminal c-Myc-tag, tolA spacer, and 3′ stem-loop without stop codon. The amplicon was purified from 1% agarose gel and used in the following cell-free protein synthesis step. We utilized RTS 100 *E. coli* HY Kit (Biotechrabbit) according to manufacturer recommendations to create the MRP complex composed of mRNA, ribosome, and protein scaffold variants.

(b) Ribosome display selection rounds. The Nunc PolySorp plate (Invitrogen, Carlsbad, CA, USA) was coated with anti-His antibody (His-tag Monoclonal Antibody (HIS.H8) by Invitrogen) diluted to a final concentration of 25 µg/mL in bicarbonate coating buffer (pH 9.6). All following steps were undertaken in the TBS buffer, pH 7.4. Blocking of plate and incubation of scaffold variants with proteins were undertaken either by an initial preselection process or an optimized preselection process described below.

(c) Initial preselection process. The plate was blocked by 3% bovine serum albumin (BSA) during all rounds. The MRP complex supplemented by 0.5% BSA and heparin (200 mg/mL) was added to the Preselection well that was coated with a mixture of 3% BSA, interferon-gamma (IFNg, 25 µg/mL), and the 4PSF wild type protein (4PSF-WT, 25 µg/mL). The reaction was incubated for 1 h at 4 °C.

(d) Optimized preselection process. The plate was blocked by either 3% BSA (used in the first and third round) or by 3% dry skimmed cow milk (used in the second and fourth round). The MRP complex supplemented by 0.5% BSA and heparin (200 mg/mL) was added to the Preselection well that was coated with a mixture of 3% BSA, IFNg (25 µg/mL), IL-29 (25 µg/mL), and 4PSF-WT (25 µg/mL). The reaction was incubated for 1 h at 4 °C.

(e) Selection process. The preselected MRP complex was transferred into the Selection well that contained IL-10 (25 µg/mL) as a target molecule and incubated again for 1 h at 4 °C. In each round, the well was washed by different cycles of washes and increasing concentration of Pluronic F-127 in TBS buffer. The conditions were as follows: First round–5 wash cycles with 0.10%, second round–10 wash cycles with 0.15%, third and fourth-round–10 wash cycles with 0.20%. The library complex was incubated with an EB buffer (50 mM Tris-Acetate, 150 mM NaCl, 50 mM EDTA, pH adjusted to 7.5) containing S. cerevisiae RNA (1 mg/mL) and heparin (200 mg/mL) to release the mRNA. The mRNA was purified by the High Pure RNA Isolation Kit (Roche, Basel, Switzerland) according to the manufacturer’s instructions and used in the GoScript Reverse Transcription System (Promega, Dane County, WI, USA) with ProBi-cloning-rev primer according to the manufacturer’s instruction. The created cDNA was amplified by PCR (supplemented by 6% DMSO) with primers ProBi-cloning-for and ProBi-cloning-rev using Q5 polymerase (New England Biolabs). The resulted amplicon was cloned into the pRDVsm vector using NcoI and BamHI restriction enzymes and T4 ligase (New England Biolabs). The pRDVsm-ProBi was used in the following preselection and selection process. After the fourth round, the amplicon was cloned into the pETsm vector and transformed into BL21(DE3)-Gold (Agilent, Santa Clara, CA, USA) for affinity estimation by enzyme-linked immunosorbent assay (ELISA) and characterization.

### 2.8. Characterization of Novel Binders

(a) ELISA assay. The Nunc PolySorp plate (Invitrogen) was coated with anti-His antibody (His-tag Monoclonal Antibody (HIS.H8) by Invitrogen) diluted to a final concentration of 5 µg/mL in bicarbonate coating buffer (pH 9.6). All following steps were done in the PBS-P buffer (PBS buffer, pH 7.4, 0.1% Pluronic F-127). For the first evaluation of the binding of 190 variants, the wells were blocked by 1% BSA, then IL-10 (10 µg/mL) was attached as a target molecule, and samples were applied in the form of cell lysates. For initial specificity mapping of 47 variants, the wells were blocked by 3% dry skim milk, 3% BSA, or 1% BSA. The wells with 1% BSA were further incubated with either IL-10 or IL-29 (10 µg/mL), and samples were applied in the form of cell lysates. For affinity estimation of 10 selected variants, the wells were blocked by 1% BSA and then incubated with IL-10 (10 µg/mL) as a target molecule. Then the samples in the form of proteins purified on StrepTactinXT were applied. Finally, the binders were detected by horseradish peroxidase (HRP)-conjugated antibody against the C-terminal c-Myc-tag and TMB-2 as substrate.

(b) Inhibitory assay. Murine RAW264.7 cells were cultured in DMEM medium supplemented with 10% FBS. The mixture containing IL-10 (final concentration of 10 ng/mL) and ProBi variants (concentrations ranging from 160 ng to 100 µg/mL) was added to the cells and incubated for 30 min. The plates with cells were placed on ice, medium discarded, and washed with a cold PBS buffer. The cells were disrupted by a cold RIPA buffer supplemented by protease and phosphatase inhibitors. An amount of 20 µg of total protein was resolved using 10% Tris-Glycine SDS-PAGE gel and transferred to nitrocellulose membrane using Trans-Blot Turbo Transfer System according to the manufacturer’s instructions. Membranes were blocked by 5% BSA in TBST buffer and incubated with antibodies against Y705-STAT3 (dilution 1/700) and alpha-tubulin (dilution 1/5000) diluted in 5% BSA in TBST buffer. Blots were washed in TBST buffer and incubated with appropriate HRP-conjugated secondary antibodies diluted in 1% BSA in TBST buffer. Membranes were treated with ECL solution (Merck Millipore, Burlington, MA, USA) and visualized on the Azure c600 instrument.

(c) Affinity measurements. The binding affinity in a solution was measured using the Monolith NT.115 instrument to monitor microscale thermophoresis (MST). The composition of the Assay buffer was 50 mM Tris, pH 8.0, 300 mM NaCl, and 0.1% Pluronic F-127. The IL-10 with C-terminal His-tag was labeled in assay buffer using the Monolith His-tag Labeling Kit RED-tris-NTA kit according to the manufacturer’s instructions. The labeled IL-10 was titrated by ProBi variants diluted in an assay buffer and incubated for 10 min at room temperature. The reaction mix was loaded into standard capillaries and MST was monitored using Medium MST power and 60% of LED power. The analysis of interaction was done in MO.Affinity Analysis v2.3 software.

(d) Circular dichroism. The folding of new ProBi variants was measured by Circular dichroism spectra the same way as done with allAla mutants of protein scaffold candidates but with a final concentration of 0.1 mg/mL.

## 3. Results and Discussion

This work aimed to formulate and test a general procedure for selecting “small engineered non-antibody protein scaffolds” [15,16] suitable for generating artificial recombinant high-affinity binders by methods of targeted in vitro evolution. The systematic procedure for the development and testing of a new stable small protein scaffold presented in this work consists of two main parts comprising the steps graphically represented in Figure 1 and summarized below:

(I) Development of a novel scaffold:

(i) Selection of suitable protein candidates from the Protein Data Bank.

(ii) In silico identification of mutable patches on the surfaces of the scaffold candidates.

(iii) Characterization of soluble wild-type variants of the scaffold candidates.

(iv) Testing the stability of the potential scaffolds.

(v) Testing the suitability of the scaffold candidate(s) for in vitro evolution.

(vi) Construction of the final ProBi scaffold.

(II) Application of the ProBi scaffold on the IL-10 target as a model system:

(i) Selection of novel ProBi-based binders by ribosome display.

(ii) Characterization of the novel ProBi-based binders.

(iii) Further development of scaffold selection and improvement of ProBi-based binders.

### 3.1. Development of a Novel Scaffold

#### 3.1.1. Selection of Suitable Protein Structures from the Protein Data Bank

The first step of the procedure was to search for appropriate structures within the Protein Data Bank (PDB) database [30] performed by the members of the laboratory. The initial search was based on visual inspection of several hundred structures of small (less than 25 kDa) monomeric proteins solved by X-ray crystallography. The search results were limited to proteins known to be expressed in *Escherichia coli*. We finally selected about 100 structures that were further evaluated based on the literature data. Our selected candidates should (i) be new protein scaffolds, (ii) be based on different folds, (iii) preferably contain no or low number of cysteines, (iv) not to be toxic, (v) not to need any cofactor to maintain the structure, and (vi) be easy to purify.

We chose 12 proteins (Table 1) for closer structural examination and a more in-depth literature review. During this step, we excluded 2W4P [48] and 4LKT [49] because of structural similarities with other selected non-human candidates, 1W2I [50] and 4I3B [51], respectively. Furthermore, we eliminated 4MJJ and 4JOX because the manuscripts were not published at that time and we could not check their expression and purification protocols. To the best of our knowledge, none of the eight finally selected proteins had previously been used as a scaffold for directed evolution.

#### 3.1.2. In Silico Identification of Mutable Surface Patches on the Protein Scaffold Candidates

On the surface of the eight candidate scaffolds, we looked for continuous patches consisting of residues that may be mutated without destabilizing the overall structure. To identify such mutable residues, we combined the analysis of protein sequence conservation with in silico saturation mutagenesis. We did not limit our search only for loop segments [28] because protein scaffolds could be mutated also at flat surfaces or combinations of loops and helices [18].

(a) Multiple sequence alignment. Sequences of protein homologs from different organisms for the eight protein scaffold candidates were aligned and the conservation of each amino acid position was calculated.

(b) In silico saturation mutagenesis. To evaluate the mutability of the residues, we performed in silico mutation scanning using the FoldX program [36]. Every amino acid position in each of the eight candidates was substituted by all of the 20 standard amino acid residues, and the corresponding free energy differences (ΔΔG) between the mutant and the wild-type structure were calculated. Thus, each position in the protein sequence was characterized by 20 ΔΔG values. We identified spots in which most of the mutations were stabilizing (ΔΔG < 0), or only slightly destabilizing (0 < ΔΔG < 0.5 kcal/mol). We calculated the mutability score for each position as a percentage of mutations fulfilling these criteria.

(c) Sequence and structural evaluation. For the selection of mutable residues, we combined the evolutionary conservation and mutability scores. Positions with identity conservation <90% and FoldX mutability score >50% were considered mutable. Next, we mapped the mutable residues onto the protein 3D structures. We visually identified mutable positions and selected those constituting compact patches on the protein surface. We preferred those patches consisting of residues that were not only structurally, but also sequentially close together, for ease of the subsequent library construction. To achieve a high variability of the designed DNA library, we selected 10–12 mutable residues, the numbers typically used in ribosome display experiments [57,58]. In cases where the patches contained less than 8–10 predicted mutable residues, we chose additional neighboring solvent-accessible residues to bring the total number of residues to 10.

In five candidate proteins, we found at least one mutable surface patch, in three of them even two independent patches (Figure 1). We excluded three proteins–4NBO [54], 3APA [55], and 4IGI [56], since they did not contain any suitable surface patch.

#### 3.1.3. Characterization of Soluble Wild-Type Variants of the Potential Scaffolds

We continued with five small protein scaffold candidates–4PSF [42], 1N3Y [52], 4I3B [51], 1W2I [50], and 2F3L [53]. We analyzed the expression level, solubility, purification simplicity, oligomerization state, secondary structure, and thermal stability of their wild-type (WT) variants.

(a) Protein expression and solubility. We prepared two versions of each scaffold candidate, first with N-terminal His-tag and second with C-terminal His-tag. We tested the expression and solubility of these five proteins in *E. coli* BL21(DE3) at four different temperatures, 37, 30, 25, and 16 °C. We detected high expression levels and excellent solubility of 4PSF, 1N3Y, 4I3B, and 2F3L with the N-terminal His-tag at 16 °C. The solubility varied with temperature. We were not able to express 4PSF, 1N3Y, and 4I3B proteins with the C-terminal His-tag. In the case of 1W2I, the proteins were not expressed in *E. coli* BL21(DE3) strain, only in *E. coli* C41(DE3) and C43(DE3) cells which are usually used for the expression of toxic proteins. Therefore, we decided to exclude the 1W2I protein.

(b) Protein purification. We continued with the purification of the remaining four proteins–4PSF, 1N3Y, 4I3B, and 2F3L, all of them as variants with the N-terminal His-tag. We performed a two-step purification procedure, first on Ni-NTA agarose beads and second by size exclusion chromatography. We isolated the 4PSF, 1N3Y, and 4I3B proteins at very high purity, but the 2F3L showed a minor band of either a homodimer or a non-relevant protein contaminant on SDS-PAGE (Figure 2A). The purified proteins retained solubility without aggregation and degradation at 4 °C in 20 mM sodium phosphate buffer at pH 7.5 and 100 mM NaCl in the range of days and weeks.

(c) Oligomerization. We used dynamic light scattering (DLS) to measure the oligomerization of purified proteins. The 4PSF, 1N3Y, and 4I3B proteins were monodisperse and monomeric based on the intensity-based particle size distribution with Z-averages of 4.6, 5.4, and 4.4 nm in diameter, respectively (Figure 2B). Although the 2F3L protein was monodisperse, the Z-average of 7.8 nm in diameter was higher than expected. It correlates with the impurities appearing on SDS-PAGE (Figure 2A). Since other than monomeric states of a protein would complicate biophysical measurements and binding assays, we decided to exclude 2F3L.

(d) Secondary structure. We checked the folding of the remaining three protein scaffold candidates, 4PSF, 1N3Y, and 4I3B, by circular dichroism (CD) measurements in the ultraviolet (UV) range (Figure 3). Based on the standard analysis of CD spectra using the CDNN software [59], the spectra of all three proteins showed features of regular secondary structures. In case of 4PSF and 1N3Y the combination of alpha-helices and beta-sheets, in case of 4I3B mainly of beta-sheets, agreeing qualitatively with the features of the respective crystal structures.

(e) Thermal stability. We determined the thermal stability of proteins by measuring their melting temperature (Tm) using a circular dichroism technique. The melting temperatures of the WT variants of 4PSF, 1N3Y, and 4I3B were 75 °C, 57 °C, and 47 °C, respectively.

#### 3.1.4. Testing the Stability of the Potential Scaffolds

Before starting a directed molecular evolution campaign, we experimentally verified the mutability of the predicted surface patches, for which we mutated all selected residues in each surface patch to alanines. The alanine-scanning mutagenesis is a method used to identify the structural role of protein residues [60,61,62]. The idea was that the protein scaffold candidate should sustain the expression, solubility, and folding upon forming a large relatively hydrophobic surface patch of alanines.

Each of three small protein scaffold candidates, 4PSF, 1N3Y, and 4I3B, has two mutable surface patches, either closer to N-terminus (PatchN) or C-terminus (PatchC) (Figure 1). Therefore, we constructed six allAla mutants. We performed the same characterization of the allAla mutants as for the WT proteins. The number of mutated residues, protein expression, solubility, and melting temperatures are summarized in Table 2. Based on these results, we decided not to continue with the 4I3B candidate because its allAla mutants did not express well or were only poorly soluble. The 1N3Y-allAla mutants were both expressed, but only the PatchN was soluble. The best results were achieved for 4PSF because both its allAla-PatchN and allAla-PatchC variants were expressed and soluble.

Next, we performed control measurements to check for non-specific binding of the 1N3Y and 4PSF scaffold candidates and their allAla mutants. We measured their binding to the capillaries used for microscale thermophoresis (MST) and to sensor chips for surface plasmon resonance (SPR) (Appendix A). We determined that the 1N3Y-WT and 1N3Y-allAla-PatchN were applicable for MST but not for SPR because they non-specifically bind to a clean surface of the SPR sensor chip. We observed no such effects with any of the 4PSF variants.

Considering all the findings, we chose the 4PSF protein as the best scaffold candidate. The WT protein as well as the allAla mutants were expressed at high yield, soluble and easy to purify, were monodispersed and folded, and possessed appropriate characteristics for MST and SPR affinity measurements. We selected the 4PSF C-terminal surface patch (PatchC) for further work because the 4PSF-PatchC-allAla mutant had the highest melting temperature among the allAla mutants.

#### 3.1.5. Testing the Suitability of the Scaffold Candidates for In Vitro Evolution

First, we constructed a degenerate DNA library for ribosome display. The selected PatchC of the 4PSF scaffold candidate had 10 calculated mutable residues (Figure 4). The random mutagenesis of these 10 positions gave rise to the maximal diversity of more than 1012 variants, which agreed with the potential diversity suitable for ribosome display [57,58]. The DNA library was synthesized by GENEWIZ company by the degenerate NNK codons technology. The same company estimated library correctness as being 78%, which was sufficient for further work.

Next, we performed pilot ribosome display selection and crystallization of variants. We took advantage of directed evolution to train the 4PSF scaffold candidate to bind the IL-10 protein as the target molecule. Pilot experiments were undertaken to establish the ribosome display protocol for the initial preselection process (see section B1 below). After the fourth selection round of ribosome display, we selected eight random colonies for two-step purification and crystallization trials. We used commercial screens for preliminary crystallization trials and then manually optimized the conditions to get better diffracting crystals. We started the crystallization with seven variants (G14, G21, H25, H33, J61, J70, and J93). We observed a crystallization process of six variants, and only the G21 did not result in any crystal form in our tested screens. Four variants created crystals with low diffraction quality (ranging 6 to 8 Å). Crystallization of the J61 variant resulted in crystals diffracting up to a high resolution (1.2 Å). We have solved the structure with the molecular replacement method using the original 4PSF structure [42]. The structure, deposited under the PDB entry ID 7AVC, was similar to the original 4PSF structure with root mean square deviation of 0.93 Å calculated on 129 Cα atoms (Appendix A). The fold remains stable even after nine mutations at the selected positions. Thus, we established the suitability of the 4PSF scaffold candidate for in vitro evolution by ribosome display. Because the affinity of the J61 variant to IL-10 was only in the micromolar range we continued in the effort to develop higher affinity binders.

#### 3.1.6. Construction of the Final ProBi Scaffold

The final 4PSF scaffold construct used for directed evolution experiments, termed ProBi-WT hereafter, consists of three sequence segments (Figure 5): (i) the N-terminal Strep-tag to simplify purification; (ii) the 4PSF scaffold segment; (iii) the C-terminal c-Myc-tag. It was added to detect variants expressed properly to the c-Myc tag and exclude fragments emerging because of premature stop codons or frameshifts in the randomized DNA library. The expression level, solubility, and biophysical features of this new ProBi scaffold construct were similar to the original scaffold candidate.

### 3.2. Application of the ProBi Scaffold on the Interleukin-10 (IL-10) Target as a Model System

#### 3.2.1. Selection of the Target Protein

Cytokines from the family of interleukin-10 are medically important signaling proteins of native immunity [31,32,33]. Immunosuppressive effects of the prototypical member of the family, IL-10, which may hinder immunotherapeutic strategies of cancer therapy [32], are not completely understood. In addition, monodispersity, CD spectra, and signaling on the RAW264.7 murine cell line of recombinant IL-10 produced in the laboratory encouraged use of this cytokine as the biologically relevant and molecularly well-defined target for our study.

#### 3.2.2. Selection of Novel ProBi-Based Binders by Ribosome Display

(a) Initial preselection process. We performed four selection rounds of ribosome display, with the following preselection conditions: incubation for an hour with a mixture of BSA, IFNg, and ProBi-WT scaffold; blocking solution 3% BSA for all rounds. After the fourth round, we selected 190 random colonies (together with ProBi-WT) for a small-scale expression in deep-96-well plates and for detection of binding to IL-10 using the ELISA assay. We took advantage of the C-terminal c-Myc-tag to reveal those clones that have the correct open reading frame and bind the IL-10. From these, we selected 47 clones with the highest signal for larger-scale expression in deep-24-well plates. These 47 clones were tested for binding to IL-29, which shares the IL-10R2 receptor with IL-10 [63]. We discovered that most of the clones showed binding to IL-29 as well as to IL-10. Most variants also displayed non-specific binding to BSA and skimmed milk.

(b) Optimized preselection process. To decrease the non-specific binding of ProBi variants to BSA and milk proteins, we optimized the preselection conditions. The initial preselection conditions were extended in three ways. We (1) included IL-29 to the preselection mix; (2) varied the blocking solution between BSA (used in the first and third round) and skimmed milk (used in the second and fourth round); and (3) performed the preselection reaction in three consecutive wells. Otherwise, we completed the ribosome display the same way as in the point a) above. The optimization of the preselection monitored on 190 variants led to a decrease of non-specific binding to the BSA and milk to 50% and 60%, respectively, of the binding to IL-10. For further characterization, we selected 10 variants (Figure 4) with high affinities to IL-10 and low affinities to IL-29, BSA, and milk. We created the phylogenetic tree that is described in Supporting Information (Appendix A).

#### 3.2.3. Characterization of Novel ProBi-Based Binders

(a) Affinity estimation by ELISA assay. We expressed and purified ten selected variants by one step purification on affinity StrepTactinXT resin and performed ELISA measurements to estimate their affinities to IL-10 and BSA (data not shown). We discarded six variants with the highest binding background to BSA and continued with four variants for more detailed characterization.

(b) Affinity measurements by MST. We expressed and purified ProBi-WT and its four variants labeled F5, G3, A2, and G6 by two-step purification on affinity StrepTactinXT resin and size exclusion chromatography. We successfully measured the IL-10 binding affinities of two variants, G3 and F5 (Figure 6). Their dissociation constants (Kd) are shown in Table 3. The F5 variant had the highest affinities, 6 nM in Tris (pH 8.0) buffer, and 15 nM in Hepes (pH 7.5) buffer, respectively. We detected binding of the A2 variant to IL-10 but we were not able to reach the bound (saturation) state of the binding curve. Therefore, we only estimated the affinities to be higher than 1 µM. The G3 variant assembled into a dimer, and the A2 and F5 variants behaved the same way as the ProBi-WT protein. The G6 variant showed the highest tendency to form oligomers (data not shown) and, therefore, it was discarded.

To evaluate the affinities of ProBi variants to IL-10, we utilized the microscale thermophoresis technique that utilizes low material consumption [64]. We used a commercial kit to fluorescently label the IL-10 with C-terminal His-tag as a target molecule and titrated it by evolved ProBi variants with the N-terminal Strep-tag. The measured affinities are shown in Table 3. We did not detect binding of IL-10 to the parental wild-type ProBi scaffold in our binding buffer (50 mM Tris-HCl, pH 8.0, 300 mM NaCl, 0.1% Pluronic F-127). We cannot measure the affinities to BSA and milk in our MST setup because they do not possess His-tag for labeling.

(c) Characterization of ProBi variants with the highest affinity. The chromatograms from size exclusion chromatography and SDS-PAGE analysis of purified proteins are shown in Appendix A. We measured the melting temperature (Tm) of the variants (Table 3) finding that Tm was lower than for the WT protein, but still in a range acceptable for practical purposes. We confirmed the folded structure of the selected ProBi variants with a high content of alpha helices according to their circular dichroism spectra (Appendix A).

(d) Inhibitory assay. We tested potential of the G3 and F5 variants to inhibit the IL-10 signaling pathway by a competitive binding assay on the RAW264.7 cell line, which expresses both IL10R1 and IL10R2 receptors on the cell surface. We used ProBi-WT protein as a negative control. Using these experimental conditions, we observed no inhibition of the IL-10 signaling pathway by either G3 or F5 (Appendix A). We hypothesize that these ProBi variants bind to the surface of IL-10 in such a way that they do not prevent IL-10 binding to the receptors IL10R1 and/or IL10R2 and do not block the signalization.

In order to identify truly inhibitory binders with a potential medical use, we need to include more ProBi variants, correlate their activity with the effect of neutralizing anti-IL-10 antibodies, test the inhibition on more cell types (e.g., THP-1 or U937), and possibly monitor signalization by outcomes other than STAT3 phosphorylation. The ProBi IL-10 binders developed as a proof of principle will have to be engineered further in order to inhibit the signaling pathway of IL-10 cytokine.

#### 3.2.4. Further Development of Scaffold Selection and Improvement of Probi-Based Binders

We are aware that the procedure of scaffold selection can be optimized, and here we discuss possible future modifications of the present protocol.

(a) The first step of our procedure, the visual screening of the PDB, can be performed more systematically. A possible way would be to automate selection of monomeric small proteins and the following identification of mutable patches on their surfaces by in silico screening by employing the FoldX [36] or Rosetta [65] programs. We can also include proteins that were produced not only in *E. coli* expression systems because the expression level and solubility of the tested scaffolds can be tested in a high-throughput format.

(b) The proposed workflow will be further tested by development of our second candidate scaffold, protein alpha-X beta2 integrin I domain (UniProt ID P20702, structure of PDB ID 1N3Y [52]) in the near future.

(c) The selectivity of the best ProBi binders to other IL-10 family cytokines was tested only by the ELISA pre-screening against IL-29 (Section 3.2.2). Further development of the binders would undoubtedly require thorough checking of their cross-reactivity to more cytokines and other proteins.

(d) A substantial challenge for possible medical use of the binders is their immunogenicity [24]. We do not expect the ProBi-based binders to be immunogenic as the scaffold is derived from the human PIH1 domain-containing protein 1 (UniProt ID Q9NWS0, structure of PDB ID 4PSF [42]). However, potentially immunogenic scaffolds could be repurposed for molecular imaging [16].

(e) The directed evolution methodologies [66,67] offer avenues to development of proteins with new properties. Over the years, several display techniques [68,69] have evolved and they offer alternative and complementary ways to the selection of optimal protein molecules for the task. In this work, we make use only one of the display techniques, ribosome display. Because the yeast display offers certain advantages compared to ribosome display, we designed the final ProBi scaffold construct to be directly used by both methods. Therefore, we included the C-terminal c-Myc-tag that is widely used for high-throughput detection in Fluorescence-activated cell sorting (FACS).

(f) The protein scaffolds as well as antibodies typically bind just one interaction partner. Two independently mutable patches on one protein scaffold molecule could open a way to train binders to interact simultaneously with two partners. Therefore, we aimed at scaffolds with two mutable patches. Such a feature would mimic a function of natural proteins, such as cytokines and/or other proteins signaling through simultaneous binding to two receptors i.e., via formation of a ternary complex. The protein scaffolds can also work as “synthekines” to engineer non-natural receptor heterodimers that could activate new unexplored cellular responses [70].

## 4. Conclusions

Protein scaffolds represent a great engineering tool that could complement functions of more commonly used antibodies in high-affinity binding of biomolecular targets. We think that there is no “one-scaffold-fits-all” and that the development of new scaffolds tailored to specific functions is crucial for real-life applications, e.g., regulation of signaling pathways in vivo, biologics, and molecular imaging.

Visual inspection of several hundred structures from the PDB, in silico investigation of about 100 selected candidates, and finally experimental examination of 12 of them, led to selection of two potentially useful scaffold proteins. The first candidate was based on the human PIH1 domain-containing protein 1 (UniProt ID Q9NWS0, structure of PDB ID 4PSF [42]) and the second one on the alpha-X beta2 integrin I domain (UniProt ID P20702, structure of PDB ID 1N3Y [52]). The latter of the two candidates is going to be developed further in the near future.

We preferred the 4PSF protein structure because it contained two surface patches amenable for independent mutagenesis. Thus, we modified the 4PSF protein construct into the protein scaffold construct called ProBi that was ready to be used in both ribosome and yeast display technology. For the purpose of this work, we focused on one of the patches and utilized the ribosome display to evolve the ProBi scaffold into the binders of interleukin-10 with nanomolar affinity. In the future, we plan to include more scaffold candidates and concentrate on their other features, such as signaling inhibition, immunogenicity, and selectivity.

In this work, we present a proof of concept methodology to identify protein structures that could be converted into a new protein scaffold constructs. We experimentally proved that at least one of them could be adapted into binders of a medically important target with nanomolar affinity by methods of directed evolution.

## Data Availability

The structural data presented in this study are openly available in the Protein Data Bank, identification number 7AVC. All other data presented in this study are available on request from the corresponding authors.

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
