# Peer review of "Protein Binder (ProBi) as a New Class of Structurally Robust Non-Antibody Protein Scaffold for Directed Evolution"

_viruses, 2021, doi:10.3390/v13020190_

Round 1

Reviewer 1 Report

In the current manuscript the authors describe a systematic process for generating small non-antibody protein scaffolds specific for human IL-10. The authors do a thorough and transparent job explaining the process, starting with in silico screening followed by cloning followed by biophysical characterization and then selection of novel binders. The authors were able to generate non-antibody protein scaffolds capable of binding IL-10 at nM affinity. Their approaches to determine binding affinities were sound and the resultant data were strong. However, the reported promise of this approach to develop binders capable of interfering with IL-10 functionality is diminished by a lack of data demonstrating this capacity in addition to a detailed explanation as to why the binders with the highest affinity failed to inhibit IL-10 signaling. Also, more explanation for the design of their inhibitory approach is requested.

  1. I strongly recommend the authors show their Y705-STAT3 blots in the manuscript. Such visual evidence is key not only for backing the author's statement that no inhibitory effect was observed but also for establishing a "baseline" for subsequent reports on future scaffold candidates.
  2. For the inhibitory assay, did the authors include a positive control to establish their ability to detect inhibition? For example, adding a known blocking anti-IL-10 antibody to their RAW cell culture system.
  3. For the inhibitory assay, being that the binders were generated to bind human IL-10 why the use of a murine cell line? A human monocytic cell line (THP-1 or U937) would seem more appropriate. Human and mouse IL-10 is reported to be 72% homologous (Moore, K.W. et al. 1990. Science 248:1230), while IL-10R homology between human and mouse is 56% (Liu, Y. et al. 1994. J. Immunol. 152:1821.). As such, I wonder if an inhibitory effect exists it may be more readily detectable using human cells.
  4. Other inhibitory assays found in routinely in the literature involve measuring the release of TNF-alpha or IL-6 by LPS-stimulated monocytic cells in the presence or absence of IL-10. As IL-10 treatment will reduce the secretion of these inflammatory cytokines, co-culture with an IL-10 inhibitor will restore the ability of the cells to secrete TNF-alpha and IL-6. Though the author's apparent finding that the current binders tested failed to suppress STAT3 activation makes the addition of another inhibitory assay to this manuscript moot, the authors may want to include this additional assay in subsequent studies testing future binders.

Author Response

Thank you for your constructive comments. Please, see our answers in the attached pdf file 

Reviewer 2 Report

1) For the selection of suitable protein structures, it is helpful to use a flow chart to demonstrate the selection and exclusion criteria and the numbers of proteins analyzed.

2) What is the affinity of the J61 variant, which the authors showed the crystal structure. 

3) The affinity of F5 variant to IL-10 is amazing. The authors should discuss why/how IL-10 was selected as a target from a technical point of view. Is the mutation on the scaffold applicable to other cytokines or a broader spectrum of targets?

4)Line-527, the authors mentioned that they performed an IL-10 inhibition assay to evaluate the G3 and F5 variant. The results should be proved regardless they are positive or negative. 

5) It will be very insightful if the authors can add more discussions on the druggability related aspects, such as immunogenicity, half-life etc. 

6) lINE 37-40 The description of antibody disadvantages should be reworded as they are not accurate. Antibody for drug development has been very successful. 

Author Response

Thank you for your constructive comments. Please, see our answers in the attached pdf file.
